# TreeSplat: Mergeable Tree for Deformable Gaussian Splatting

**Qiuhong Shen**[1]    **Xingyi Yang**[2,1]    **Xinchao Wang**[1*]

[1]National University of Singapore    [2]The Hong Kong Polytechnic University

qiuhong.shen@u.nus.edu, xingyi.yang@polyu.edu.hk, xinchao@nus.edu.sg

## Abstract

Dynamic 3D scene reconstruction from multi-view videos demands representation to model complex deformations at scale. Current Gaussian Splatting based methods often either suffer from significant computation cost due to dense MLP-based modeling or explicit modeling deformation of each Gaussian independently. However, the dynamics of objects within a scene are typically hierarchical and exhibit structural correlations. To leverage these structural priors into the representation, we introduce **TreeSplat**, a **Tree** data structure for deformable Gaussian **Splat**ting. In TreeSplat, as the name suggests, motions of Gaussian are represented hierarchically within a tree. Each node learns coefficients for time-varying basis functions, defining a part of the motion. The full motion for any given Gaussian is then determined by accumulating these transformations along the tree path from its leaf node to the root node. This tree isn't predefined; instead, it is constructed adaptively alongside Gaussian densification, where cloning or splitting a Gaussian correspondingly creates new leaf nodes. One central property of TreeSplat is its mergeability; after optimization during training, the hierarchical motion parameters for each Gaussian can be efficiently consolidated. By performing this merging step before test time, we eliminate the need to traverse the tree explicitly for each Gaussian during rendering. This results in dramatically faster rendering over 200 FPS and compact storage, while maintaining state-of-the-art rendering quality. Experiments on diverse synthetic and real-world datasets validate these advantages. Code and results are available at `https://github.com/florinshen/treesplat`.

## 1   Introduction

Dynamic 3D scene reconstruction aims to recover deformable 3D representations from multi-view videos. This, in turn, makes it possible to synthesize novel views from different viewpoints and moments in time. Despite recent progress, achieving efficient dynamic 3D scene reconstruction remains challenging due to limitations in fitting time, data storage, and rendering speed.

Neural Radiance Fields (NeRF) [1–6] have demonstrated remarkable performance in static 3D reconstruction by optimizing multi layer perceptrons (MLPs) with volumetric ray marching. Several extensions [7–9, 9–18] incorporated temporal components into the MLP to capture scene deformations over time. However, these methods often suffer from high computational demands, significantly impacting their efficiency, particularly when scaling to complex dynamic scenes.

Recent developments in 3D Gaussian Splatting (3DGS) [19] have established it as a compelling alternative for efficient 3D scene reconstruction. Building upon this foundation, a growing body of work [16, 20–37] has incorporated temporal components into static Gaussians to model scene deformations over time. Among them, some approaches adopt isotropic 4D Gaussians [22, 23] or

---

[*]Corresponding Author.

39th Conference on Neural Information Processing Systems (NeurIPS 2025).

polynomial functions [24–26] to explicitly represent motion. These representations circumvent dense MLP computation and thus enable efficient rendering.

Despite these success, a key limitation of these methods is the *independent* modeling of motion for each Gaussian. In reality, the dynamics are not independent. They are inherently correlated across space and time, often exhibiting complex, hierarchical structures. Consequently, assuming independence prevents the capture of coherent and structured motion patterns, creating a significant modeling gap. Thus, it is crucial to model deformations collaboratively, allowing Gaussians to share information with their neighbors and better reflect the structured nature of real-world motion.

To achieve this, we introduce **TreeSplat**, a novel representation that employs a tree data structure to explicitly model both individual Gaussian motions and their inter-correlations. The core idea is to organize motion hierarchically within this tree. Specifically, each node stores coefficients for a set of shared, time-varying basis functions, with these coefficients defining a component of the motion. The complete motion for any Gaussian is then determined by aggregating these component transformations—typically via weighted summation—along the path from its associated leaf node to the tree's root. Consequently, the tree explicitly defines the motions for all Gaussians, and crucially, Gaussians that share common ancestor nodes inherently share motion components, thus modeling their correlation.

One central problem is how we construct the tree. Rather than manual or rule-based creation, we adaptively grow the tree in conjunction with Gaussian densification. As standard densification operations, cloning and splitting generate new Gaussians in adjacent regions, the tree structure grows alongside them.

Specifically, we define two forms tree growth modes, that operate alternately. *Leaf Expansion* broadens the tree. It links motion of new Gaussians (resulting from densification) with new leaf nodes under the same parent as their source Gaussian. *Depth Promotion* increases the depth of the tree. It transforms an existing Gaussian's leaf node into a new parent, which then links it to two new child leaf nodes associated with newly generated Gaussians. Furthermore, to control the influence of different tree levels, each Gaussian applies a learnable decay to the coefficients of motion nodes with increasing depth. This ensures that motion components defined by nodes deeper in the tree have a smaller impact, while components closer to the root dictate more significant.

Our motion tree offers a key advantage: its coefficients are **mergeable** after training. This mergeability means that although the tree structure is essential for training, it is not required at inference time. Once training is complete, we can merge the coefficient for each Gaussian by traversing its leaf-to-root path and accumulating the corresponding transformations. Crucially, this merging process occurs *offline*, before test time. Consequently, we eliminate the requirement for per-Gaussian tree traversal and significantly accelerating runtime performance.

To maintain computational efficiency during training, we periodically prune inactive motion nodes, preserving a sparse and lightweight tree structure. Once training converges, the tree structure is fixed, and the motion coefficients are merged offline as described. This final representation, with pre-aggregated motion for each Gaussian, enables highly efficient rendering of complex dynamic scenes without runtime tree traversal overhead.

In summary, the contributions of this work are as follows:
• We propose a hierarchical motion tree structure for dynamic Gaussian Splatting, where the tree is grown in conjunction with Gaussian densification to model collaborative and structured motion.
• The motion tree is formulated as mergeable, in which the linear structure enables offline pre-aggregation of motion coefficients. This design eliminates the need for tree traversal during rendering, thereby substantially improving efficiency and reducing storage overhead.
• Our TreeSplat framework achieves state-of-the-art reconstruction quality on both synthetic and real-world datasets, while maintaining real-time rendering speed over 200 FPS.

## 2   Related works

**Dynamic 3D Scene Reconstruction.** 3D Gaussian Splatting (3DGS) [19, 38–40] have become a key approach for efficient 3D scene reconstruction. A stream of recent works [16, 20–36, 41–43] has extended 3DGS to dynamic scene reconstruction by introducing temporal components to model deformations over time. These methods can be broadly divided into two categories based on how they

incorporate temporal components: implicit and explicit deformation modeling. In implicit approaches, several representative methods [20, 21] employ a shared MLP to predict the deformation of each Gaussian over time. This design is similar to deformation components used in dynamic NeRFs. However, real-world dynamic scenes often require millions of Gaussians for accurate reconstruction. As a result, querying an MLP for each Gaussian introduces significant computational overhead, which becomes a major bottleneck during rendering. To mitigate this issue, more recent work has focused on explicit deformation representations that avoid per-Gaussian MLP inference. Such methods either encode deformation using isotropic 4D Gaussians [22, 23] or parameterize the deformation of each Gaussian using polynomial functions of time [24–26]. While these techniques improve efficiency in both training and rendering, they learn motion parameters independently for each Gaussian and ignore the spatial dependencies between motions of Gaussians. This independence assumption neglects the hierarchical and coherent motion patterns often present in dynamic scenes, leaving a critical modeling gap. In contrast, our approach introduces a hierarchical and mergeable motion tree that explicitly models collaborative motion across Gaussians, enabling improved dynamic scene reconstruction and rendering effiency.

**Hierarchy in Gaussian Splatting.** Hierarchical representations are fundamental in 3D reconstruction. Early works [44, 45] introduced strategies such as spatial partitioning and pyramid features into NeRF [1] to improve both rendering speed and visual quality across diverse 3D scenes. Thanks to its fully explicit formulation, Gaussian Splatting allows for more direct modeling over spatial structure, making it a natural candidate for hierarchical representation. Several recent works [46–50] have explored constructing hierarchical representations based on Gaussian Splatting. HierarchicalGS [46] proposed a tree-based hierarchy that enables the reconstruction of large-scale scenes by dividing them into independent spatial chunks. OctreeGS [48] employed an octree data structure to adaptively capture level-of-detail variations across different scene regions, while SVRaster [47] also adopted an octree structure and introduced a sparse voxel rasterization method to mitigate popping artifacts [51] in Gaussian Splatting rendering.

More recently, HiCoM [49] explored hierarchical motion modeling using 4DGaussian [20] for streamable dynamic scene reconstruction. However, its hierarchy is constructed using fixed heuristics, which limits adaptability across varying scenes. Concurrent work HiMoR [50] introduced a fixed-structure tree of depth 3 to represent motion for monocular dynamic scene reconstruction. Different from these approaches, our framework focuses on mining motion hierarchies between Gaussians in an adaptive manner. We couple tree construction with the densification process, enabling the tree to grow adaptively in both depth and width to reflect motion complexity. Moreover, our linear tree structure supports pre-aggregation of motion coefficients after training, enabling highly efficient rendering without additional tree traversal overhead.

## 3 Preliminary: 3D Gaussian Splatting

Given a set of images captured from 3D scenes with known camera poses, 3D Gaussian Splatting (3DGS) [19] iteratively reconstructs the scene as a collection of isotropic Gaussians, denoted as $\{\mathcal{G}\}$. Each Gaussian is defined by its central position $\boldsymbol{\mu} \in \mathbb{R}^3$, opacity $\sigma \in [0, 1]$, and view-dependent color $\boldsymbol{h} \in \mathbb{R}^{48}$ represented as spherical harmonics coefficients. Its 3D covariance matrix $\Sigma^{3D} \in \mathbb{R}^{3 \times 3}$ is parameterized by a scale vector $\boldsymbol{s} \in \mathbb{R}^3$ and a rotation quaternion $\boldsymbol{q} \in \mathbb{R}^4$ to ensure positive semi-definiteness. Collectively, the $i$-th Gaussian is denoted as $\mathcal{G}_i = \{\boldsymbol{\mu}_i, \sigma_i, \boldsymbol{h}_i, \boldsymbol{s}_i, \boldsymbol{q}_i\}$.

To better represent the scene, adaptive density control, or densification, is employed by dynamically adjusting the number of Gaussians during optimization. It selectively densifies regions with large view-space positional gradients, which typically indicate either missing geometry (under-reconstruction) or excessive coverage (over-reconstruction). Specifically, small Gaussians in under-reconstructed regions are cloned to cover new geometry, while large Gaussians in high-variance areas are split into smaller Gaussians to capture finer scene details.

For rendering a 2D image, all 3D Gaussians are first projected onto the image plane as 2D Gaussians. The opacity of Gaussian $i$ at a pixel $\mathbf{x}$ is given by:

$$\alpha = \sigma_i \exp\left(-\frac{1}{2}(\mathbf{x} - \bar{\boldsymbol{\mu}}_i)^T \Sigma_i^{2D^{-1}} (\mathbf{x} - \bar{\boldsymbol{\mu}}_i)\right), \tag{1}$$

where $\bar{\boldsymbol{\mu}}_i$ and $\Sigma_i^{2D}$ are the projected mean and covariance, respectively. After sorting Gaussians based on depth, a tile-based rasterizer is used to blend their contributions, producing the final image.

# 4 Methodology

In this section, we first establish the base formulation of dynamic Gaussian Splatting in Sec. 4.1. The detailed data structure of the motion tree is elaborated in Sec. 4.2, and the tree merging strategy for rendering is described in Sec. 4.3. Finally, the initialization and optimization scheme for dynamic Gaussian Splatting with the motion tree is presented in Sec. 4.4.

## 4.1 Dynamic Gaussian Splatting

To adapt 3D Gaussian Splatting for dynamic scene reconstruction, we adopt explicit temporal components to capture deformations over time. Specifically, per-Gaussian coefficients associated with shared basis function are applied to encode temporal motion and rotation of each Gaussian as shown in Fig. 1(a). Rather than relying on predefined motion bases such as Fourier series [24] or polynomial [26] basis, we initially represent the motion field through per-Gaussian coefficients [25] with learnable bases:

$$\boldsymbol{\mu}(t) = \boldsymbol{\mu}_c + \hat{\boldsymbol{\mu}}(t) = \boldsymbol{\mu}_c + \sum_{j=1}^{B} c_j^{\mu} \boldsymbol{b}_j^{\mu}(t), \quad \boldsymbol{q}(t) = \boldsymbol{q}_c + \hat{\boldsymbol{q}}(t) = \boldsymbol{q}_c + \sum_{j=1}^{B} c_j^{q} \boldsymbol{b}_j^{q}(t), \qquad (2)$$

where $\boldsymbol{\mu}_c$ and $\boldsymbol{q}_c$ are the static Gaussian center and rotation quaternion, respectively, and $B$ denotes the number of basis functions. Here, $c_j^{\mu} \in \mathbb{R}$ and $c_j^{q} \in \mathbb{R}$ are learnable coefficients. The basis functions $\boldsymbol{b}_j^{\mu}(t) \in \mathbb{R}^3$ and $\boldsymbol{b}_j^{q}(t) \in \mathbb{R}^4$ are shared across all Gaussians and are generated by a lightweight multi-layer perceptron (MLP) $f_\theta$ with a sinusoidal time embedding:

$$f_\theta(\gamma(t)) = \{(\boldsymbol{b}_j^{\mu}(t), \boldsymbol{b}_j^{q}(t))\}_{j=1}^{B}, \quad \gamma(t) = \left(\sin(2^k \pi t), \cos(2^k \pi t)\right)_{k=0}^{L-1}, \qquad (3)$$

where $\theta$ denotes the learnable parameters of the MLP, and $\gamma(t)$ is a sinusoidal positional encoding of order $L$ to enhance the network's capacity to model high-frequency temporal variations.

Compared to MLP-based deformation modeling that requires querying the MLP millions of times [20, 21], once for each Gaussian, our framework only performs a single query to $f_\theta$ per timestamp $t$, substantially reducing the computational overhead.

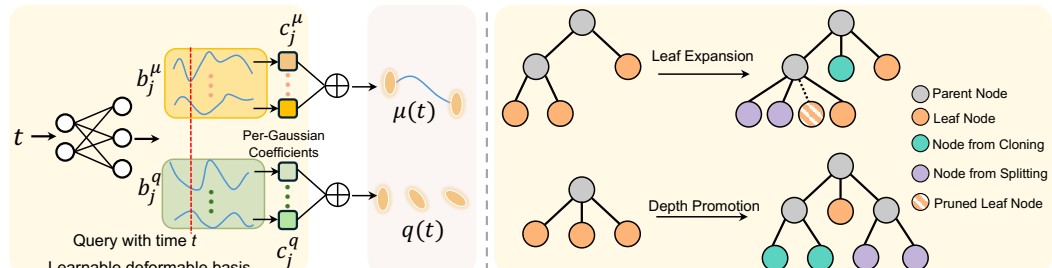

(a) Dynamic Gaussian with Learnable Basis     (b) Leaf Expansion and Depth Promotion

Figure 1: **Dynamic Gaussian Splatting with Hierarchical Motion Tree.** Our framework encodes temporal motion and rotation using shared learnable basis functions combined with per-Gaussian coefficients. Each node in the hierarchical motion tree stores a motion vector parameterized by learnable coefficients. The tree is constructed in conjunction with Gaussian densification, where *Leaf Expansion* and *Depth Promotion* are performed periodically to adaptively grow the hierarchy.

## 4.2 Hierarchical Motion Tree

Though the above dynamic Gaussian Splatting models per-Gaussian deformation using shared basis functions, the associated coefficients are optimized independently for each Gaussian. However, motions in real-world 3D scenes are inherently spatially entangled and exhibit hierarchical relationships across multiple scales, where the motion of one structure is often correlated with the motions of its neighbors. Motivated by this property, we introduce a hierarchical motion tree to collaboratively learn the temporal motion $\hat{\boldsymbol{\mu}}(t)$ in Eq. 2 among Gaussians.

In the motion tree structure, each node represents a motion vector encoded as a learnable coefficient vector in $\mathbb{R}^B$, and the entire motion field is denoted as $\mathcal{F} \in \mathbb{R}^{M \times B}$, where $M$ is the total number of

nodes in the tree. Each leaf node is uniquely associated with one Gaussian, ensuring a one-to-one correspondence between Gaussians and leaf nodes. Given $N$ ($N \leq M$) dynamic Gaussians, the structure of the motion tree is maintained by the following data mappings:

- Node entry $\mathcal{E} \in \mathbb{N}^N$: maps each Gaussian to its leaf node index in the motion field $\mathcal{F}$;
- Parent index $\mathcal{P} \in \mathbb{N}^M$: records the parent motion node for each motion node;
- Node level $\mathscr{L} \in \mathbb{N}^N$: specifies the number of nodes from Gaussian's leaf node to the root node;

Given a constructed motion tree, the motion $\hat{\boldsymbol{\mu}}_i(t)$ of Gaussian $i$ is computed by traversing its ancestral path with parent index $\mathcal{P}$ and aggregating motion vectors. Formally, we can formulate as:

$$\hat{\boldsymbol{\mu}}_i(t) = \sum_{k=0}^{\ell_i} \sum_{j=1}^{B} \mathcal{F}_j^{\mathrm{anc}(i,k)} \boldsymbol{b}_j^\mu(t), \tag{4}$$

where $\ell_i$ denotes the level of Gaussian $i$ in the tree, and $k$ indexes the nodes along its path from the leaf to the root. The function $\mathrm{anc}(i,k)$ returns the index in $\mathcal{F}$ of the ancestor node $k$ steps above the leaf node of Gaussian $i$.

**Tree Growth: Leaf Expansion and Depth Promotion.** As Gaussian densification progressively refines the scene from coarse to fine spatial levels by creating new Gaussians in neighboring regions, it naturally aligns with the hierarchical motion modeling motivation. The growth of the motion tree is in conjunction with the densification during optimization. To adaptively balance the depth and width of the motion tree, we introduce two alternative modes to grow the tree at each densification step as shown in Fig. 1(b).

*Leaf Expansion* increases the number of leaf nodes horizontally without increasing tree depth. Newly densified Gaussians are linked to new leaf nodes under the same parent node as their source Gaussian. These new leaf nodes share the same parent index $p \in \mathbb{N}$ and node level $l \in \mathbb{N}$ as the leaf node of their source Gaussian, preserving the flatness of the local tree structure.

*Depth Promotion* deepens the tree to refine the hierarchical organization of motions. It transforms an existing Gaussian's leaf node into a new parent node. The densification operation, whether cloning or splitting, is treated as a binary operation that creates two new leaf nodes associated with Gaussians and connect them to the newly promoted parent node. These new leaf nodes are assigned a node level of $l+1$ and their parent indices $p$ point to the newly promoted parent node.

For all newly created leaf nodes, the associated coefficients in the motion field $\mathcal{F}$ are initialized to zero. This initialization ensures that the aggregated temporal motion $\hat{\boldsymbol{\mu}}(t)$ remains unchanged immediately before and after densification.

This two growth modes are scheduled periodically during optimization: multiple rounds of Leaf Expansion are followed by a Depth Promotion step. This design results in a large, adaptively widening motion tree with limited depth, ensuring that the tree remains both scalable and capable of capturing multi-level motion structures throughout training. Besides, unused leaf nodes are periodically remove during optimization to maintain efficiency during training.

**Decay-Weighted Aggregation.** As the motion tree deepens, naive aggregation of ancestor nodes, as formulated in Eq. 4, can result in unstable optimization. Specifically, given the upstream per-Gaussian gradient $\boldsymbol{g}_i = \partial \mathcal{L} / \partial \hat{\boldsymbol{\mu}}_i$, the gradient with respect to an ancestor node $\mathcal{F}^{\mathrm{anc}(i,k)}$ is computed as

$$\frac{\partial \mathcal{L}}{\partial \mathcal{F}^{\mathrm{anc}(i,k)}} = \left( \boldsymbol{g}_i^\top \boldsymbol{b}_1^\mu(t), \, \boldsymbol{g}_i^\top \boldsymbol{b}_2^\mu(t), \, \ldots, \, \boldsymbol{g}_i^\top \boldsymbol{b}_B^\mu(t) \right) \in \mathbb{R}^B \tag{5}$$

In this naive aggregation, ancestor node accumulates the full strength of the upstream gradient from each of its descendants. Consequently, the total gradient magnitude at a node grows proportionally to the number of its descendant Gaussians, which can cause severe gradient explosion for some ancestor nodes and destabilize optimization.

To mitigate this issue, we introduce a learnable decay factor $\beta_i \in (0, 1]$ for each Gaussian. The motion aggregation is reformulated as:

$$\hat{\boldsymbol{\mu}}_i(t) = \sum_{k=0}^{\ell_i} \left( \prod_{p=0}^{k-1} \beta_i \right) \sum_{j=1}^{B} \mathcal{F}_j^{\mathrm{anc}(i,k)} \boldsymbol{b}_j^\mu(t), \tag{6}$$

where deeper ancestors are exponentially attenuated by powers of $\beta_i$. Correspondingly, gradients received by each ancestor node is scaled by the same decay factor, ensuring that gradients from distant descendants are substantially suppressed. The decay factors $\beta_i$ are differentiable and jointly optimized with other model parameters during training. A closed-form derivation of their gradients is provided in the *Appendix*.

This *decay-weighted aggregation* strategy enables each Gaussian to adaptively control its receptive field within the motion tree: a smaller $\beta_i$ concentrates motion learning on nearby ancestors, while a larger $\beta_i$ permits broader aggregation. By suppressing contributions from distant and irrelevant ancestors, it improves the robustness of both forward motion modeling and backward gradient flow, leading to more stable motion modeling.

### 4.3 Tree Merging for Rendering

During optimization, computing the motion $\hat{\boldsymbol{\mu}}_i(t)$ for each Gaussian requires traversing its ancestral path in the deformation tree. Once training is complete and the tree structure is fixed, we pre-aggregate the motion coefficients for each Gaussian by reordering the summation terms in Eq. 6. Formally, it can be expressed as:

$$\hat{\boldsymbol{\mu}}_i(t) = \sum_{j=1}^{B} \boldsymbol{b}_j^{\mu}(t) \sum_{k=0}^{\ell_i} \left( \prod_{p=0}^{k-1} \beta_i \right) \mathcal{F}_j^{\mathrm{anc}(i,k)} = \sum_{j=1}^{B} \boldsymbol{b}_j^{\mu}(t) \boldsymbol{c}_j^i \tag{7}$$

where $\boldsymbol{c}^i \in \mathbb{R}^B$ is the merged motion coefficient vector for Gaussian $i$. This transformation enables highly efficient rendering, as it eliminates the need to traverse the tree, the motion computation reduces to simple linear combination over the $B$ basis vectors per Gaussian.

It is important to note that this coefficient merging is only applied after training. During optimization, the internal motion nodes must be retained to receive gradient signals from multiple descendant Gaussians and propagate them back across the tree. Merging would prematurely remove the hierarchical structure necessary for effective gradient flow and motion disentanglement. Hence, mergeability is a property leveraged only in the inference stage to accelerate rendering without compromising the training dynamics.

### 4.4 Training Framework

Following 3D Gaussian Splatting [19], we conduct interleaved iterative optimization and densification. Since the tree construction process is coupled with densification, the time overhead introduced by growing the deformation tree is negligible. Moreover, with our highly optimized CUDA implementation, tree traversal for each Gaussian during optimization introduces only a marginal time increase.

**Tree Initialization.** We first use point cloud to initialize $N_{\mathrm{init}}$ Gaussians. The node levels $\mathscr{L}$ are initialized to 0 for all Gaussians, corresponding to the root level of the tree. The initial motion field $\mathcal{F}$ has the same length as the number of Gaussians, with all entries initialized to zero. For the node entry mapping $\mathcal{E}$, each Gaussian is assigned to its corresponding motion node in order, initialized as $[0, 1, \ldots, N_{\mathrm{init}} - 1]$. The parent indices $\mathcal{P}$ are initialized to $-1$ for all nodes, forming a forest of independent singleton trees at initialization.

**Optimization Scheme.** During training, we simply adopt a rendering loss composed of an MSE term and an SSIM term: $\mathcal{L} = (1 - \lambda)\mathcal{L}_{\mathrm{mse}} + \lambda\mathcal{L}_{\mathrm{ssim}}$, where $\lambda$ balances the two objectives. To stabilize training, we initially reconstruct the static components of $\boldsymbol{\mu}(t)$ and $\boldsymbol{q}(t)$ over the first 10% of optimization iterations, effectively learning a canonical 3D space before introducing temporal dynamics. For interleaved densification, we adopt the averaged screen-space 2D gradient of $\boldsymbol{\mu}(t)$ as the density control indicator.

## 5 Experiments

### 5.1 Implementation details

To balance efficiency and performance, we model the time-varying shared basis $f_\theta$ in Eq. 3 using a MLP with three hidden layers, each of width 512. The sinusoidal time embedding $\gamma(t)$ is set

as order $L = 32$ to capture high-frequency temporal variations. The number of basis vectors $B$ is set to 10 for the D-NeRF dataset and 16 for the Neural 3D Video dataset, to accommodate the complexity of real-world scenes. Motion tree construction begins at iteration 500 in conjunction with Gaussian densification. *Depth Promotion* is performed at first step, after which *Leaf Expansion* is applied every 100 iterations and *Depth Promotion* every 500 iterations, alternating after every four rounds of *Leaf Expansion*. Densification is halted at iteration 15,000, after which the motion tree is fixed to capture stable collaborative motion patterns among Gaussians. To ensure fast per-frame motion computation, we implement Eq. 6 and Eq. 7 as custom CUDA kernels separately. Through careful kernel design, including memory access optimization and parallel thread scheduling, we ensure that tree traversal overhead is negligible even for real-world dynamic scenes. All other training hyperparameters, including densification thresholds and learning rates for Gaussian attributes, follow the original settings used in Gaussian Splatting. The learning rate for the decay factor $\beta$ is set to $5 \times 10^{-3}$, and the learning rate for the shared MLP $f_\theta$ is set to $5 \times 10^{-4}$. All experiments are conducted on an NVIDIA RTX4500 Ada GPU. For additional hyperparameter configurations, please refer to the configuration table provided in the *Appendix*.

## 5.2 Comparison with State-of-the-art

**D-NeRF Dataset [7].** The D-NeRF dataset comprises 8 synthetic dynamic scenes captured as monocular videos. At each time step, only a single training image from one viewpoint is available. Following standard protocols, we evaluate on test views from novel camera positions within the same temporal range as the training data. For initialization, we uniformly sample 100,000 points within the cubic volume $[-1.2, 1.2]^3$. Quantitative results are reported in Tab. 1 in terms of PSNR, SSIM, and LPIPS. Our TreeSplat achieves the highest reconstruction quality with an average PSNR of 37.11 dB, while maintaining a compact model size of 28 MB and requiring only 4 minutes of training per scene. After tree merging, TreeSplat reaches a rendering speed of 230 FPS, outperforming all baselines. Moreover, it uses significantly fewer Gaussians, averaging only 85K per scene, demonstrating superior efficiency without compromising quality.

Table 1: Quantitative Evaluation on the D-NeRF [7] Dataset.

| Method | PSNR↑ | SSIM↑ | LPIPS↓ | Train Time↓ | FPS↑ | Storage↓ | #Gauss↓ |
|---|---|---|---|---|---|---|---|
| DNeRF [7] | 29.67 | 0.95 | 0.08 | 40 h | 0.1 | N/A | N/A |
| K-Planes [52] | 31.07 | 0.97 | 0.02 | 5 h | 1.2 | N/A | N/A |
| HexPlanes [8] | 31.04 | 0.97 | 0.04 | 11 min | 0.22 | N/A | N/A |
| TiNeuVox [53] | 32.67 | 0.97 | 0.04 | 49 min | 1.6 | N/A | N/A |
| 4DGaussian [20] | 34.05 | 0.98 | 0.02 | 19 min | 85 | 19M | 137K |
| CompactDGS [24] | 32.19 | 0.97 | 0.04 | 4 min | 202 | 32M | 112K |
| DynMF [25] | 35.72 | 0.98 | 0.02 | 5 min | 216 | 32M | 101K |
| 4DRotorGS [22] | 34.26 | 0.97 | 0.03 | 26 min | 143 | 242M | 392K |
| RealTime4DGS [23] | 34.09 | 0.98 | 0.02 | 28 min | 132 | 278M | 445K |
| TreeSplat (ours) | 37.11 | 0.98 | 0.02 | 4 min | 230 | 28M | 85K |

**Neural 3D Video Dataset.** The Neural 3D Video (N3V) dataset consists of 6 indoor dynamic scenes captured with 18 to 21 cameras at a resolution of $2704 \times 2028$, each lasting ten seconds. Following standard protocols, we train and evaluate at the half resolution, using 300 frames per scene with the center camera held out for evaluation. For initialization, we sample 300,000 points from the COLMAP [54] point cloud using farthest point sampling [55]. Tab. 2 reports quantitative results in terms of PSNR, storage size, rendering speed, and training time. We compare our method against 7 NeRF-based approaches and 5 representative Gaussian Splatting methods. Among them, 4DGaussian [20] and Grid4D [35] employ MLP-based modeling, while DynMF [25], RealTime4DGS [23], and SpaceTimeGS [26] adopt explicit deformation representations. Our TreeSplat achieves the highest average reconstruction quality of 32.21 dB across scenes, while maintaining a compact model size of 170 MB and the fastest rendering speed of 206 FPS after tree merging. Remarkably, TreeSplat also trains significantly faster than most baselines, requiring only 0.57 hours on average per scene. Besides, we qualitatively compare novel view synthesis results on two representative scenes (*Coffee Martini* and *Sear Steak*) in Fig. 2, showing our method yields better reconstruction. These results highlight TreeSplat's superior trade-off between reconstruction quality and efficiency.

Table 2: Quantitative Evaluation on the Neural 3D Video [12] Dataset.

| Method | PSNR (dB) | | | | | | | MB | Frame/s | Hours |
|---|---|---|---|---|---|---|---|---|---|---|
| | Coffee Martini | Cook Spinach | Cut Roasted Beef | Flame Salmon | Flame Steak | Sear Steak | Average | Size | FPS | Training time |
| NeRFPlayer [13] | 31.53 | 30.56 | 29.35 | 31.65 | 31.93 | 29.13 | 30.69 | 5130 | 0.05 | 6 |
| HyperReel [14] | 28.37 | 32.30 | 32.92 | 28.26 | 32.20 | 32.57 | 31.10 | 360 | 2 | 9 |
| DyNeRF [12] | N/A | N/A | N/A | 29.58 | N/A | 29.58 | 29.58 | 28 | 0.015 | 1344 |
| HexPlane [8] | N/A | 32.04 | 32.55 | 29.47 | 32.08 | 32.39 | 31.71 | 200 | N/A | 12 |
| K-Planes [8] | 29.99 | 32.60 | 31.82 | 30.44 | 32.38 | 32.52 | 31.63 | 311 | 0.3 | 1.8 |
| MixVoxels-L [15] | 29.63 | 32.25 | 32.40 | 29.81 | 31.83 | 32.10 | 31.34 | 500 | 37.7 | 1.3 |
| MixVoxels-X [15] | 30.39 | 32.31 | 32.63 | 30.60 | 32.10 | 32.33 | 31.73 | 500 | 4.6 | N/A |
| 4DGaussian [20] | 27.34 | 32.46 | 32.90 | 29.20 | 32.51 | 32.49 | 31.15 | 34 | 137 | 1.7 |
| Grid4D [35] | 28.30 | 32.58 | 33.22 | 29.12 | 32.56 | 33.16 | 31.49 | 146 | 116 | 1.9 |
| DynMF [25] | 28.87 | 33.09 | 32.66 | 29.03 | 32.70 | 32.02 | 31.40 | 176 | 197 | 0.52 |
| RealTime4DGS [23] | 28.33 | 32.93 | 33.85 | 29.38 | 34.03 | 33.51 | 32.01 | 2085 | 101 | 5.2 |
| SpaceTimeGS [26] | 28.61 | 33.18 | 33.52 | 29.48 | 33.64 | 33.89 | 32.05 | 200 | 140 | 5.5 |
| TreeSplat (ours) | 28.91 | 33.17 | 33.69 | 29.50 | 33.89 | 34.10 | 32.21 | 170 | 206 | 0.57 |

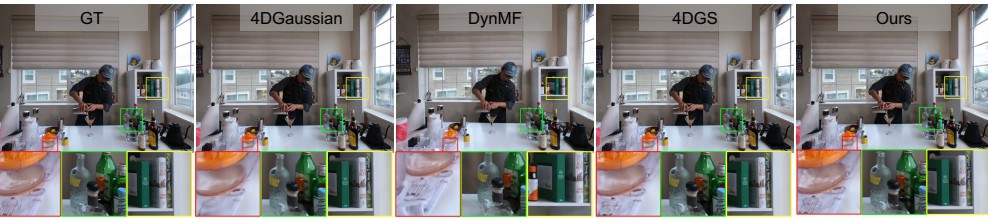

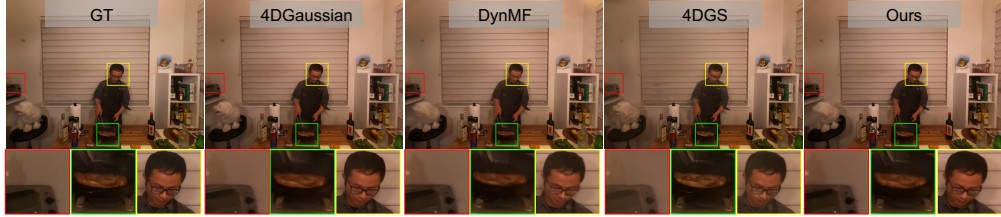

Figure 2: **Qualitative comparisons on the Neural 3D Video Dataset.** We qualitatively compare our TreeSplat with 4DGaussian [20], DynMF [25], and RealTime4DGS [23] on two representative scenes *coffee martini* and *sear steak* from the Neural 3D Video dataset [12].

## 5.3 Ablation and Discussion

**Effects of Tree Depth.** Our hierarchical motion tree grows in conjunction with Gaussian densification through periodically scheduled *Leaf Expansion* and *Depth Promotion*. The overall tree depth is controlled by adjusting the interval between *Depth Promotion* steps. In Tab. 3, we conduct an ablation study across five configurations with maximum tree depths of 0, 4, 15, 29, and 73, corresponding to *Depth Promotion* intervals of $\infty$, 4000, 1000, 500, and 200, respectively. The results show that performance consistently improves as tree depth increases. Compared to the baseline without a hierarchical structure (tree depth 0), our full model with depth 29 achieves notable PSNR gains, such as $+2.63$ on *Hellwarrior* and $+1.40$dB on *Sear Steak*. Even moderate depths (e.g., 4 or 15) already yield substantial improvements, demonstrating that our tree structure captures meaningful motion correlations with limited hierarchy.

Table 3: **Ablation of tree depth.** Evaluated on representative scenes from D-NeRF (*Hellwarrior*, *Mutant*) and Neural 3D Video (*Cut Roasted Beef*, *Sear Steak*).

| Tree Depth | Hellwarrior | | Mutant | | Cut Roasted Beef | | Sear Steak | |
|---|---|---|---|---|---|---|---|---|
| | PSNR↑ | SSIM↑ | PSNR↑ | SSIM↑ | PSNR↑ | SSIM↑ | PSNR↑ | SSIM↑ |
| 73 | 38.26 | 0.97 | 43.28 | 1.00 | 33.31 | 0.96 | 34.05 | 0.97 |
| 29 | 38.34 | 0.97 | 43.81 | 1.00 | 33.69 | 0.96 | 34.10 | 0.97 |
| 15 | 38.11 | 0.97 | 42.57 | 1.00 | 32.79 | 0.96 | 33.81 | 0.97 |
| 4 | 37.49 | 0.97 | 42.79 | 1.00 | 33.63 | 0.96 | 33.39 | 0.96 |
| 0 | 35.71 | 0.95 | 41.47 | 0.99 | 32.15 | 0.96 | 32.65 | 0.96 |

However, excessively deep trees can reduce the width of the tree and may accumulate irrelevant motion noise. Moreover, they introduce greater computational overhead and may even cause minor performance degradation. Empirically, we find that a moderate depth of 29 (interval 500) offers the best balance between performance, efficiency, and robustness.

**Effects of Tree Merging.** The main difference between the training-time motion computation in Eq. 6 and the rendering-time formulation in Eq. 7 lies in the order of aggregation. During training, we first compute time-dependent motion vectors for all nodes in the motion tree and then aggregate them for each Gaussian. In contrast, rendering pre-aggregates the motion coefficients per Gaussian before applying the basis functions, thereby eliminating the need to traverse the tree. While these two formulations are mathematically equivalent, numerical differences can arise due to floating-point accumulation order. To evaluate the impact of such differences, we conduct ablation experiments comparing models with and without coefficient merging. As shown in Tab. 4, merging introduces negligible variation in reconstruction quality across PSNR, SSIM, and LPIPS, while significantly improving rendering speed and reducing storage size. These results validate the effectiveness of our mergeable design, which preserves visual fidelity while enabling high-performance inference.

Table 4: **Ablation on Tree Merging.** We report the metric difference as "after merging" minus "before merging." Merging preserves reconstruction quality while improving efficiency and compactness.

| Scene | $\Delta$PSNR | $\Delta$SSIM | $\Delta$LPIPS | $\Delta$Storage (MB) | $\Delta$FPS |
|---|---|---|---|---|---|
| Hellwarrior | $-3.81 \times 10^{-7}$ | $+5.96 \times 10^{-9}$ | $-4.47 \times 10^{-8}$ | $-0.62$ | $+15$ |
| Mutant | $+9.53 \times 10^{-7}$ | $+2.98 \times 10^{-9}$ | $-1.97 \times 10^{-8}$ | $-1.55$ | $+12$ |
| Cut Roasted Beef | $+3.81 \times 10^{-8}$ | $-7.94 \times 10^{-10}$ | $-7.45 \times 10^{-9}$ | $-8.57$ | $+82$ |
| Sear Steak | $-1.27 \times 10^{-8}$ | $+1.39 \times 10^{-9}$ | $+7.45 \times 10^{-9}$ | $-8.61$ | $+74$ |

**Effects of Weight-Decayed Aggregation.** To assess the impact of weight-decayed aggregation, we conduct ablation by disabling it during training on the D-NeRF dataset. As illustrated in Fig. 3, removing this strategy leads to blurred motion regions in the rendered results. This is caused by interference from irrelevant nodes in the motion tree. In contrast, enabling this strategy yields much better reconstruction quality, as the learnable decay factor $\beta$ in Eq. 6 adaptively controls each Gaussian receptive field in the motion tree.

| Scene | w/o Window | | With Window | |
|---|---|---|---|---|
| | PSNR | #Gauss | PSNR | #Gauss |
| Hellwarrior | 38.34 | 19K | 37.87 | 22K |
| Mutant | 43.81 | 70K | 41.35 | 79K |
| Cut roasted beef | 33.69 | 416K | 33.36 | 460K |
| Sear steak | 34.10 | 429K | 33.89 | 519K |

Table 5: **Impact of Temporal Opacity window.** We compare with and without opacity window.

Figure 3: **Ablation of weight-decayed aggregation.**

**Analysis of Tree Depths.** To better understand the structure of motion hierarchies, we analyze the distribution of tree depths across four representative scenes: *Hellwarrior* and *Mutant* from the DNeRF dataset, and *Cut Roasted Beef* and *Sear Steak* from the Neural 3D Video dataset. As shown in Figure 4, we divide tree depths into six intervals and visualize the results as histograms. For each interval, we also report the average number of Gaussians per tree. The resulting distributions exhibit a long-tailed pattern: over 97% of motion trees have depths less than or equal to 10, while only a small fraction extend into deeper hierarchies. These findings demonstrate that most Gaussians are captured through relatively shallow hierarchical motion patterns, with deeper structures emerging only where necessary.

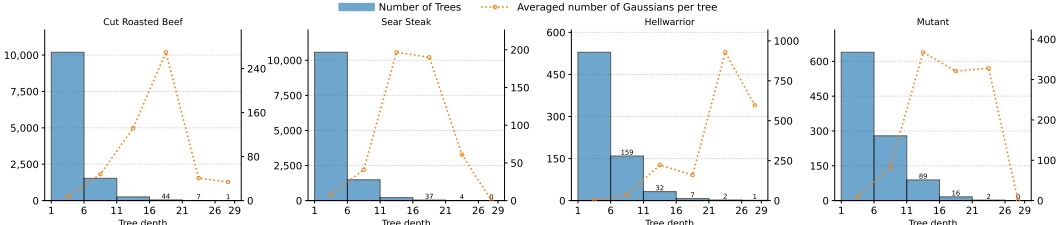

Figure 4: **Visualization of hierarchical tree statistics.** Here we show statistics across four dynamic 3D scenes (*Cut Roasted Beef*, *Sear Steak*, *Hellwarrior*, and *Mutant*). For each scene, the histogram (blue bars) represents the number of trees at different depth intervals, while the dotted orange curve denotes the averaged number of Gaussians per tree within each depth range.

**Temporal Opacity.** Some prior works [23, 26] model the opacity $\sigma$ of each Gaussian as a time-varying unimodal function, often implemented as a temporal window. However, recent findings [56, 57] suggest that Gaussians tend to learn overly short temporal windows, resulting in limited lifespans and degrading dynamic 3D reconstruction into near per-frame static modeling. To evaluate the necessity of temporal opacity in our framework, we follow the same design and assign each Gaussian a learnable 1D Gaussian window. As reported in Tab. 5, introducing temporal windows degrades reconstruction quality despite a noticeable increase in the number of Gaussians. This likely occurs because Gaussians with limited temporal visibility hinder the motion tree from learning coherent motion trajectories. These results confirm that temporal windowing is unnecessary in our framework. The temporal modulation of opacity can be sufficiently expressed through Gaussian center motion, as reflected in the spatial formulation of opacity in Eq. 1.

## 6 Conclusion

In this work, we present a hierarchical and mergeable motion tree structure for dynamic Gaussian Splatting. By constructing motion tree in conjunction with Gaussian densification, our method enables collaborative motion learning across Gaussians in an adaptive, coarse-to-fine manner. A decay-weighted aggregation strategy further regulates the influence of ancestor nodes, improving both optimization stability and motion locality. To support efficient rendering, we introduced a pre-aggregation strategy that merges motion coefficients over the tree path of each Gaussian, eliminating traversal overhead during rendering. Extensive experiments on both synthetic and real-world datasets demonstrate that our method achieves state-of-the-art reconstruction quality while maintaining high efficiency and compactness. Our approach offers a principled framework for hierarchical motion modeling and sets the stage for future extensions in scalable dynamic 3D representations.

## Acknowledgement

This project is supported by the National Research Foundation, Singapore, under its Medium Sized Center for Advanced Robotics Technology Innovation.

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
