# OpenReview forum: "TreeSplat: Mergeable Tree for Deformable Gaussian Splatting"
_NeurIPS.cc/2025/Conference — NeurIPS 2025 poster_

### Official Review · Reviewer_wnrg · 2025-06-10

**Clarity:** 3
**Significance:** 3
**Originality:** 2
**Rating:** 4
**Confidence:** 3

**Summary:**

Overall, this submission focuses on the Dynamic 3D scene reconstruction task. Instead of regarding each Gaussian kernel to hold independent motion patterns, this submission aggregate the motion patterns of correlated Gaussian kernels to better simulate the real-world scenario and thus achieve better performance in general.

**Questions:**

Overall, the reviewer currently holds around a borderline view for this submission. Below are his current concerns.

1.  Firstly, the authors seem to deeply couple the construction of the tree with the densification process of the original Gaussian Splatting pipeline. The reviewer holds the following concerns w.r.t. this design choice.

1.1. On the one hand, this choice seems to assume that any Gaussians densified from the same parent Gaussian must be correlated (i.e., share the same motion pattern group). The reviewer is curious w.r.t. this assumption. For example, while two Gaussians cloned from the same one can be close to each other once they are just cloned, it is possible that they are learnt to become generally distant from each other during the optimization process. In this case, is it reasonable or limiting to assume that potentially distant Gaussians to necessarily be under the same motion pattern?

1.2. Moreover, the proposed method seems to lack mechanisms that can be used to split the tree into two or more different trees as well. The reviewer, especially with the above concern 1.1, is hesitating on whether this counts a limitation and thus appreciates further clarification and explanation.

2. The authors are appreciated to better elaborate their difference from existing works, e.g., [48]. For example, when the authors say that [48] only use a fixed-structure tree to represent motion. Does this mean [48]'s mechanism choose to do so or [48]'s mechanism only supports pre-fixed mechanism. The reviewer believes that such difference from existing works, especially the most similar few, needs to be better elaborated. Meanwhile, the emprical comparison with such very similar methods, such as [47, 48] is also felt to be necessary by the reviewer.

3. W.r.t. the experiments part, the reviewer is confused w.r.t. the reported training time. Specifically, if the reviewer is not wrong, the authors even use the whole section 4.3 to show how they accelerate rendering during the inference stage. This leaves the reviewer an expectation that the training may be not very fast for this method. Yet, referring to the experiments section, the reviewer surprisingly find that the training time is very fast. The authors are appreciated to explain this part. E.g., do they train the pipeline for the same number of iterations as existing methods?

4. Meanwhile, it is good to see that the authors admit that their method "brings moderate
overhead during training". Yet, as discussed in the above concern 3, this overhead seems to be not from the time perspective but rather from the memory perspective? If this is the case, the authors are appreciated to report such overhead beyond just the Storage memory (Model size) in Table 1 and Table 2, in order for the readers to better get the characteristics of this method.

**Ethical Concerns:**

["NO or VERY MINOR ethics concerns only"]

**Final Justification:**

After reading the author's rebuttal, I think my concerns are largely solved. Thus, despite the admitted increase in GPU memory overhead, I keep my positive rating of this submission.

**Limitations:**

Yes

**Paper Formatting Concerns:**

N.A.

**Quality:**

3

**Strengths And Weaknesses:**

Strength:

1. Overall, this submission is easy-to-understand.

2. The emphrical results of this submission in general demonstrate its effectiveness.

Weakness:

(See questions section for more details)

---

> ### Author Rebuttal · Authors · 2025-07-30
>
> Thank you for your insightful comments! Below, we provide a point-by-point response to address your concerns. We welcome further discussion to improve the clarity and effectiveness of our work.
>
> > **Q1-1: Correlation between Gaussians from same parent.**
>
> We thank the reviewer for this insightful observation. It is indeed possible for two Gaussians cloned from the same parent to gradually diverge in motion trajectory during optimization. Our model addresses this through the decay-weighted motion aggregation defined in Eq. (6), where each Gaussian inherits motion from its ancestors scaled by a learnable decay factor $\beta$. Empirically, we observe that the learned $\beta$ values after training span a wide range: from as low as $3.64 \times 10^{-9} $ to 1.0, with a mean of 0.15. When $\beta_i$ approaches zero, the influence from parent motion nodes becomes negligible, allowing the Gaussian $G_i$ to rely primarily on its own leaf motion. This mechanism provides flexibility for Gaussians that were initially correlated (due to cloning) to learn independently when required, effectively address this limitation.
>
> > **Q1-2:  Lack mechanisms to split the tree into two or more different trees as well.**
>
> We appreciate the reviewer’s thoughtful question. While our method does not include an explicit tree-splitting mechanism, the decay-weighted motion aggregation introduced in Eq. (6) implicitly enables flexible motion decoupling. Specifically, since the influence of ancestor nodes is attenuated exponentially by powers of the learned decay factor $\beta$, each Gaussian can adaptively learn to minimize its reliance on earlier parts of the tree. When the accumulated decay becomes negligible (i.e., $\beta^k \approx 0$ ), the Gaussian effectively becomes disconnected from its ancestor nodes. This in practice achieves the effect of tree splitting without requiring explicit structural modifications.
>
> > **Q2: Comparison with similar works of fixed tree structure.**
>
> We thank the reviewer for highlighting the need for a more detailed comparison with prior hierarchical motion modeling works. HiMoR [48] adopts a fixed 3-layer tree structure that is manually defined and not adaptable during training. This reflects a design constraint rather than a design choice, HiMoR’s architecture only supports pre-defined hierarchies. In contrast, our method builds a dynamic motion tree structure in a data-driven manner.
>
> Direct empirical comparison with these two methods is challenging, as HiMoR [48] is tailored for monocular dynamic scene reconstruction and Hicom [47] is developed for streamable dynamic scenes. Both are evaluated on domain-specific benchmarks that differ from ours. To enable a fair comparison, we re-implemented their respective hierarchical motion strategies within our framework. As shown in the table below, our learned adaptive tree consistently outperforms these fixed or heuristic hierarchies across two representative scenes, demonstrating the effectiveness of our approach.
>
> |      Method      | PSNR (Cut Roasted Beef) | SSIM | PSNR (Sear Steak) | SSIM |
> | :--------------: | :---------------------: | :--: | :---------------: | :--: |
> | HiMoR-reproduced |          32.68          | 0.96 |       32.25       | 0.96 |
> | Hicom-reproduced |          31.93          | 0.95 |       32.11       | 0.96 |
> | Treesplat (ours) |          33.69          | 0.96 |       34.10       | 0.97 |
>
> We will clarify these differences in the related work section and include the corresponding empirical results in the revised paper.
>
> > **Q3: Confusion about training efficiency.**
>
> We apologize for the confusion. Our pipeline uses the same number of iterations (30k) as existing dynamic Gaussian Splatting works. Given that the method involves a large number of Gaussians with varying node levels, we devoted substantial engineering efforts to implement both the forward and backward passes of Eq.~(6) as optimized CUDA kernels (more details can be found in Appendix B.1). This ensures high computational efficiency during training. As a result, the additional overhead introduced by the hierarchical tree structure is minimal: compared to our direct baseline DynMF, the average training time on the Neural 3D Video dataset only increases from 31 minutes to 34 minutes. We will clarify this by explicitly mentioning these speedup details at the last paragraph of Section 4.2 in the revision.
>
> > **Q4: Overhead of motion tree compared to baselines.**
>
> We appreciate the reviewer’s suggestion. As mentioned in the response of Q3, the extra training time overhead of our method is minimal. However, we acknowledge that the GPU memory overhead  is moderately increased due to additional data structures required to maintain the hierarchical motion trees during training. Specifically, our method incurs an average of 200MB additional memory on DNeRF scenes and 850MB on Neural 3D Video scenes. We will include GPU memory usage during training in Tables 1 and 2 in the revised version. This will provide a more complete view of the method’s resource characteristics.

---

> > ### Comment · Reviewer_wnrg · 2025-08-07
> >
> > Hi authors,
> >
> > I am sorry that previously, I thought you can see the final justification and I thus left my feedback to your rebuttal there.
> >
> > As mentioned there, after reading the author's rebuttal, I think my concerns are largely solved. Thus, despite the admitted increase in GPU memory overhead, I keep my positive rating of this submission.

---

> > > ### Author Response · Authors · 2025-08-07
> > >
> > > Dear Reviewer wnrg,
> > >
> > > Thank you very much for your considerate follow up message and for letting us know that our rebuttal has addressed your concerns. We appreciate your continued positive assessment of our work.
> > >
> > > In our revised manuscript we will add a clearer discussion of the observed increase in GPU memory usage so that readers can better understand the associated tradeoffs.
> > >
> > > If any additional questions arise, we would be delighted to address them.
> > >
> > > With sincere appreciation,
> > >
> > > The Authors of Submission 2287

---

### Official Review · Reviewer_gmM8 · 2025-06-27

**Clarity:** 2
**Significance:** 2
**Originality:** 2
**Rating:** 4
**Confidence:** 4

**Summary:**

The paper introduces TreeSplat, a Gaussian Splatting-based framework for dynamic 3D scene reconstruction. The core idea is to structure Gaussians hierarchically into trees, enabling deformation to be defined at the tree level with learnable coefficients. The tree hierarchy is constructed using the splitting and cloning strategies from 3D Gaussian Splatting (3DGS), resulting in a forest structure. This design allows MLP computations to be applied at the tree level rather than per Gaussian, reducing computational overhead, while maintaining reconstruction quality.

**Questions:**

1. As I understand it, the number of root nodes is dependent on the number of points in the initial point cloud. Since the root node count likely has the largest impact on computational overhead, is there any analysis between the size of the initial point cloud and performance?

2. How much time does the tree merging process take in practice?

3. Could you clarify the method used to identify the "unused leaf nodes" mentioned in weakness 3?

4. I'm curious about how Gaussians are distributed within each tree. For example, do Gaussians within the same tree tend to represent the same object, or is there no such correlation?

5. What are the SSIM and LPIPS results on the Neural 3D Video dataset?

6. Overall, what is the typical decay factor between the root and leaf nodes in the tree?

**Ethical Concerns:**

["NO or VERY MINOR ethics concerns only"]

**Final Justification:**

After carefully reading the authors' reply and other reviewers' comments, I have decided my final rating as borderline accept. My main concern is related to the visual results presented by the authors. Specifically, the current results shown in the main paper and supplementary materials are insufficient to clearly demonstrate how the proposed method (representing Gaussians using a tree structure) contributes to the improvements. I believe this issue has been partially addressed through the authors' clarification and the additional materials being prepared. Aside from this issue, I agree with the novelty of the proposed tree construction mechanism. I recommend that the authors further include results demonstrating the proposed tree mechanism and its improvements.

**Limitations:**

As noted in the weaknesses, the paper lacks sufficient analysis of the proposed model and does not present results that clearly align with the authors' claims, making it difficult to assess the actual effectiveness of the approach.

**Quality:**

2

**Strengths And Weaknesses:**

### Strengths
1. This paper proposes a method for constructing a hierarchy of Gaussians without relying on heuristics, enabling motion hierarchy modeling that differs from prior approaches.

2. The tree structure is built using the splitting and pruning from 3DGS, which is both intuitive and effective.

3. The method demonstrates improved performance over existing approaches in quantitative evaluations.

### Weaknesses
1. In qualitative video results, moving objects (e.g., head of a dog or the hand) appear noticeably blurred, making it difficult to assess where rendering quality has actually improved.

2. The paper lacks side-by-side video comparisons with other methods, which makes it hard to verify the claimed improvements visually.

3. Some implementation details are missing. For instance, the term "unused leaf nodes" mentioned in line 202 is not clearly defined.

4. The paper lacks an in-depth analysis of the tree structure's formation and behavior. Since computational overhead likely scales with the number of root nodes, it would be helpful to analyze the size and structure of the tree/forest in different scenes.

5. It appears that the purpose of modeling a motion hierarchy is to automatically capture dependencies among the motions of Gaussians. Therefore, a detailed comparison with heuristic motion hierarchy generation methods, and a clearer analysis of how the proposed tree structure differs and performs better, would be valuable. This is not sufficiently addressed.

6. If deeper trees lead to better results, it implies that Gaussians are more independent. However, this seems misaligned with the paper's claim that capturing coherent motion is better.

7. I wonder why comparisons with D-3DGS[21] and SC-GS[32] on the D-NeRF dataset are missing.

8. The term tree merging is confusing. It is used to describe precomputing motion coefficients within a single tree, rather than merging multiple trees. A clearer terminology would help avoid misunderstanding.

---

> ### Author Rebuttal · Authors · 2025-07-30
>
> Thank you for your insightful comments! Below, we provide a point-by-point response to address your concerns. We welcome further discussion to improve the clarity and effectiveness of our work.
>
> > **W1: Motion blur in video results.**
>
> We acknowledge this is one of our limitations. Such blurred artifacts are commonly observed in dynamic Gaussian Splatting without incoporating the temporal opacity (as described in Sec 5.3). Through ablation experiments, we found that enabling temporal opacity helps reduce such motion blur artifacts, particularly on fast-moving parts like the head of the dog. However, it introduces a notable increase in the number of Gaussians and slightly compromise overall rendering quality. We believe this reflects a trade-off between spatial sharpness and temporal consistency.
>
> > **W2: Side-by-side video comparison.**
>
> We appreciate the reviewer’s interest in comparison with video results. While videos are not able to be included in the rebuttal, we kindly invite the reviewer to refer to Fig.2, which presents a side-by-side visual comparison with three representative baselines: 4DGaussian, DynMF (our direct baseline), and 4DGS. Visual improvements are zoomed in and cropped for clarity. Our method consistently produces sharper details and fewer artifacts, supporting the improvements we claim.
>
> > **W3 & Q3: Clarification on unused leaf nodes.**
>
> Thank you for pointing this out. The term unused leaf node refers to those leaf motion nodes whose associated Gaussians have been pruned during training. We will clarify this term in the revision.
>
> > **W4: Tree structure and computation overhead analysis.**
>
> Thank you for the insightful comment. In practice, our implementation is parallelized at the Gaussian level. Specifically, our custom CUDA kernel launches a thread for each Gaussian, which independently traverses its associated motion tree path to compute the aggregated motion. Therefore, the actual computational overhead scales with the number of Gaussians, and the computation time (latency) is influenced by the maximum depth of the tree path each Gaussian must traverse. For the tree structure, we analyze the distribution of tree depths across scenes. As shown in the table presented in response to reviewer QfP4, the tree structures follow a long-tailed distribution, with over 97% of motion trees exhibiting depths within 10. This indicates that the majority of Gaussians are connected to shallow trees, ensuring efficient computation while still capturing hierarchical motion.
>
> > **W5: Compare to heuristic motion hierarchy.**
>
> Thank you for the reviewer’s suggestion to include comparisons with heuristic motion hierarchy construction.  We implemented a baseline on the Neural 3D Video dataset by generating heuristic hierarchies using semantic segmentation. Specifically, the SfM points from initialization are projected onto the first frame of Camera 0. We apply the Segment Anything Model (SAM) to segment this frame and assign each projected point a semantic label based on its 2D location. Points projected outside the image are labeled as invalid (−1). For each valid semantic label, we manually construct a two-layer motion tree, assigning a shared parent motion node and a leaf node to each Gaussian within the same segment. Experimental results on two representative scenes (Cut Roasted Beef and Sear Steak) show that this heuristic tree structure leads to a drop in reconstruction quality.
> | Method           | PSNR (Cut Roasted Beef) | SSIM | PSNR (Sear Steak ) | SSIM |
> | :---------------: | :---------------------: | :--: | :----------------: | :--: |
> | Heuristic        |          32.73          | 0.96 |       32.69       | 0.96 |
> | Treesplat (ours) |          33.69          | 0.96 |       34.10        | 0.97 |
>
> Such heuristic motion hierarchy faces difficulty in reliably inferring motion dependencies from sparse SfM points in dynamic scenes, where inaccurate correlations could introduce significant noise and hinder reconstruction. In contrast, our tree is adaptively constructed in conjunction with Gaussian densification, guided solely by reconstruction loss.
>
> > **W6: Clarification on tree depth vs. motion coherence.**
>
> We appreciate the reviewer’s insightful concern. As shown in the table presented in response to reviewer QfP4, most Gaussians are connected to shallow trees, indicating local motion coherence rather than independence. Moreover, our ablation experiments of tree depth (from line 315 to 325) in Sec. 5.3 demonstrate that moderate tree depth leads to the best reconstruction performance, deeper trees do not necessarily yield better results. This suggests that the benefits of TreeSplat come from modeling local hierarchies, rather than from increasing depth arbitrarily.
>
> > **W7: Comparison with D-3DGS [21] and SC-GS[32].**
>
> We apologize for this omission. This was due to differences in the experimental protocols: all dynamic Gaussian Splatting baselines in Table 1 were trained with 30k iterations for comparison, whereas D-3DGS and SC-GS were trained with 40k and 80k iterations respectively. Including them directly may introduce inconsistency.  Given the reviewer’s helpful suggestion, we will include both methods in revision, with a clear table footnote clarifying this discrepancy to ensure informative comparison.
>
> > **W8: Potential confusion of the term Merging.**
>
> We appreciate the reviewer’s comment regarding this potential confusion.  *Tree merging* refers to merge coefficients rather than merging the tree structure. It is the process of consolidating the hierarchical motion coefficients along a Gaussian's ancestral path into a single coefficient vector prior to rendering. We agree that the current term may be confusing and will adopt clearer terminology such as motion coefficient consolidation in the revision to better reflect the actual process.
>
> > **Q1: Impact of Initial Point Cloud Size on Performance.**
>
> We thank the reviewer for this insightful question. We have empirically validated the impact of initial point cloud size on performance. On the DNeRF dataset, the initialization points are randomly sampled from a cubic volume. Most of these points are pruned before tree construction (typically by the 500th iteration), resulting in minimal influence on the final motion tree structure and negligible effect on runtime.
>
> On the Neural 3D Video dataset, initialization is based on SfM point clouds, sampled using farthest point sampling. We varied the number of initialization points from the baseline 300k to 600k and 30k. Increasing from 300k to 600k leads to no observable training time increase, and the resulting number of Gaussians remains in the same order of magnitude. However, reducing the initialization to 30k increases the average training time from 34 minutes to 41 minutes. This is because a sparser initialization requires the model to densify more Gaussians during training, which in turn leads to a larger number of deep motion trees.
>
> As discussed in our response to W4, the computation time (latency) is influenced by the maximum depth of the tree path each Gaussian must traverse. With a larger number of deep trees, each CUDA block during kernel launch is more likely to encounter threads with deep tree paths, leading to an observable increase in training time.
>
> > **Q2: Time cost of tree merging.**
>
> Thank you for the question. The tree merging process is highly efficient with our customized CUDA kernel. It takes 0.84 ms per scene on DNeRF and 1.22 ms per scene on Neural 3D Video dataset when evaluated on NVIDIA RTX4500 Ada GPU. This is negligible compared to the training time and has no impact on rendering quality. We will clarify this point in the revised version.
>
> > **Q4: Internal structure of motion trees.**
>
> We appreciate the reviewer’s interest in the internal structure of the motion trees. To investigate this, we visualized Gaussian trajectories on the Neural 3D Video dataset by coloring all Gaussians bound to the same motion tree with same color. The results reveal that coherent motion trees consistently emerge around moving objects. For instance, in the flame_salmon scene, a motion tree clearly corresponds to the motion of a hand, with its associated Gaussians localized around the hand. In contrast, the static background is typically composed of a large number of isolated Gaussians without parent node (i.e., node level = 0). This behavior arises naturally from our tree growth mechanism, which is coupled with Gaussian densification and driven by gradients from reconstruction loss. Since moving regions introduce higher pixel inconsistencies across time, their corresponding gradients tend to be larger, promoting more active tree growth and hierarchical modeling in dynamic regions.
>
> > **Q5: SSIM and LPIPS on N3V Dataset.**
>
> We sincerely apologize for omitting them in the manuscript. In the following table, we provide the per-scene SSIM and LPIPS results. These results will be included in the revised version.
>
> |      Scene       | PSNR  | SSIM | LPIPS |    Scene     | PSNR  | SSIM | LPIPS |
> | :--------------: | :---: | :--: | :---: | :----------: | :---: | :--: | :---: |
> |  Coffee Martini  | 28.91 | 0.92 | 0.061 | Cook Spinach | 33.17 | 0.96 | 0.042 |
> | Cut Roasted Beef | 33.69 | 0.96 | 0.044 | Flame Salmon | 29.50 | 0.93 | 0.056 |
> |   Flame Steak    | 33.89 | 0.97 | 0.032 |  Sear Steak  | 34.10 | 0.97 | 0.035 |
>
> > **Q6:  Typical decay factor.**
>
> We thank the reviewer for this question. For the learned decay factor $\beta$, we computed statistics across all Gaussians associated with leaf motion nodes on Neural 3D Video dataset. Initialized at 0.5, the learned $\beta$ values after training range from a minimum of $3.64 \times 10^{-9}$ to a maximum of 1.0, with a mean value of 0.15. This indicates that while some Gaussians retain strong influence from ancestors, many exhibit rapid decay, enabling flexible control of motion inheritance depth.

---

> > ### Comment · Reviewer_gmM8 · 2025-08-04
> >
> > I appreciate the authors' thoughtful response. Additionally, I have further questions that I would like to ask.
> >
> > I need further clarification on the pruning operation described in W3 & Q3. Did the authors directly apply the pruning strategy from 3DGS, or was another pruning operation introduced?
> >
> > Also, when initializing the neural 3D video dataset, did the authors use points from all timestamps? If points from different timestamps were indeed used, was farthest point sampling consistently applied?

---

> > > ### Author Response · Authors · 2025-08-04
> > >
> > > We appreciate reviewer's follow-up questions and provide further clarifications below.
> > >
> > > > **Clarification on pruning.**
> > >
> > > Thanks for pointing out these details. Our method does not introduce any additional pruning operation. We adopt the same pruning strategy as 3DGS and follow the same pruning threshold setting as defined in the original 3DGS implementation. This detail will be included in the revised version together with the definition of unused leaf nodes.
> > >
> > > > **Details on SfM initialization on the N3V dataset.**
> > >
> > > We also thank the reviewer for raising this point. During initialization, we only use points estimated from the first timestamp across all training cameras, following the common practice in prior dynamic Gaussian Splatting works [20, 22, 23]. Accordingly, farthest point sampling is applied only to the SfM points estimated from this first timestamp.

---

> > > > ### Author Response · Authors · 2025-08-06
> > > >
> > > > Dear Reviewer gmM8,
> > > >
> > > > We wish to convey our sincere appreciation for your insightful feedback, which has been of great help to us. As the discussion deadline approaches, we are keenly anticipating any additional comments or suggestions you may have. Ensuring that the rebuttal aligns with your suggestions is of utmost importance. We are deeply grateful for your commitment to the review process and your generous support throughout.
> > > >
> > > > Once again, thank you for your valuable time and sound advice!
> > > >
> > > > Best regards,
> > > >
> > > > The Authors of Submission 2287

---

> > > > > ### Comment · Reviewer_gmM8 · 2025-08-06
> > > > >
> > > > > Dear Authors,
> > > > >
> > > > > I have no further questions to raise. The authors have adequately addressed my concerns. Indeed, most of my concerns relate to visual results, which I understand cannot be provided within this rebuttal; I will take this into consideration.

---

> > > > > > ### Author Response · Authors · 2025-08-06
> > > > > >
> > > > > > Dear Reviewer gmM8,
> > > > > >
> > > > > > Thank you for confirming that our rebuttal has addressed your concerns. We appreciate your understanding regarding the visual results. More qualitative figures will be included in revised version.
> > > > > >
> > > > > > Your careful review and constructive comments have greatly improved the clarity of our work. Should any further questions arise, please feel free to let us know.
> > > > > >
> > > > > > With sincere appreciation,
> > > > > >
> > > > > > The Authors of Submission 2287

---

> > > > > > ### Author Response · Authors · 2025-08-09
> > > > > >
> > > > > > Dear Reviewer gmM8,
> > > > > >
> > > > > > Thank you for the constructive and thoughtful discussion. Based on our exchange, we understand the main remaining concern is about the visual results. For the final version, we will add several qualitative results we have already prepared. These will include (1) visualizations of the learned motion tree structures to show how they capture coherent motion (as discussed in Q4), and (2) more detailed, cropped comparisons against baselines to better highlight the improvements in sharpness and artifact reduction.
> > > > > >
> > > > > > The discussion window closes in about four hours. If any point would benefit from a brief textual clarification, we can respond promptly before the window closes.
> > > > > >
> > > > > > We would greatly appreciate your final evaluation, as it will provide valuable guidance for further improving our paper before finalization.
> > > > > >
> > > > > > Best regards,
> > > > > >
> > > > > > The Authors of Submission 2287

---

### Official Review · Reviewer_QfP4 · 2025-07-01

**Clarity:** 3
**Significance:** 3
**Originality:** 4
**Rating:** 5
**Confidence:** 4

**Summary:**

This paper introduces TreeSplat, a novel method for dynamic 3D scene reconstruction based on Gaussian Splatting. The key contribution is a hierarchical motion tree structure designed to model the correlated and structured movements inherent in dynamic scenes, addressing the limitation of prior works that model each Gaussian's motion independently. In this framework, the motion of any given Gaussian is determined by accumulating transformations along a path from its associated leaf node to the tree's root. Critically, this tree structure is not predefined but grows adaptively in conjunction with the standard Gaussian densification process, forming a "forest" of trees that represent different moving parts of the scene. The paper's most significant technical innovation is the "mergeability" of this tree; after training, the hierarchical motion parameters can be pre-aggregated into a single coefficient vector for each Gaussian. This eliminates the need for tree traversal during inference, enabling state-of-the-art reconstruction quality at real-time rendering speeds (over 200 FPS).

**Questions:**

Potential for User Control: Your learned tree provides a powerful hierarchical motion representation. Have you considered its potential for user interaction, similar to the control offered by anchor-based methods like SC-GS? For instance, could manipulating a high-level node in your tree (e.g., a learned "shoulder" node) allow for intuitive control over all its descendants (the "arm" Gaussians)? While we understand this is not the paper's primary goal and do not expect experimental results, a brief discussion on this potential would be a valuable addition to contextualize your work.
Robustness to Topological Changes & Role of Tree Growth: The paper does not seem to address how the model handles scenes with significant topological changes (e.g., an object breaking into two pieces). The learned tree establishes a fixed parent-child relationship for the duration of the training. Could you clarify how the model would represent the motion of two diverging pieces that were once connected? Am I correct in understanding that the "Leaf Expansion" and "Depth Promotion" mechanisms are designed solely to build and refine the motion hierarchy, rather than to adapt to such topological changes by, for instance, splitting a tree into two? A clarification on the precise role and limitations of these mechanisms would be very helpful.
Analysis of the Learned Forest Structure: As mentioned before, to better understand the model's behavior, could you provide some analysis of the learned forest structure? For example, a simple table with statistics on the number of trees and their average/max depth for different scenes would be very insightful. Furthermore, a qualitative visualization coloring the Gaussians by their tree ID would be highly compelling evidence that the method correctly segments objects based on their correlated motion.

**Ethical Concerns:**

["NO or VERY MINOR ethics concerns only"]

**Final Justification:**

After I check all discuss during rebuttal, I respect author's efforts and based on all information I got, I give my final score as before.

**Limitations:**

The paper would benefit from a dedicated "Limitations" section. The authors have presented a strong conclusion, and adding a formal discussion of the method's boundaries would create a more complete and well-rounded manuscript. I suggest this section could acknowledge and reflect upon some of the points raised in the "Weaknesses" and "Questions" parts of this review.

**Quality:**

3

**Strengths And Weaknesses:**

Strengths:
1.	The paper tackles a well-defined and significant problem in dynamic scene reconstruction: modeling coherent, structured motion efficiently. The core idea of an adaptively grown, mergeable motion tree is highly original. It shifts the paradigm from modeling independent primitives or relying on implicit MLPs to automatically discovering and representing the physical hierarchy of motion.
2.	The method is technically sound and its effectiveness is demonstrated through outstanding results. It achieves state-of-the-art performance on both synthetic (D-NeRF) and Neural 3D Video datasets. The ability to combine top-tier reconstruction quality with extremely fast rendering speeds (>200 FPS).
3.	The technical design is elegant. Coupling the motion tree's growth with the Gaussian densification process is a clever and natural way to build the motion hierarchy. Furthermore, the "mergeable" design, which separates the complex training structure from a simple and fast inference model, is a crucial insight for practical application. The paper's quality is further reinforced by a comprehensive set of ablation studies that convincingly validate key design choices
Weakness
1.	The paper's core contribution is the motion forest, yet the experiments lack a perspective of this emergent structure itself. The paper does not provide quantitative data (e.g., the number and size distribution of trees per scene) or direct qualitative visualizations (e.g., coloring Gaussians by their tree ID) to explicitly show how the learned trees map to distinct, correlated objects. Such analysis would provide invaluable insight into the method's inner workings and offer stronger, more direct evidence for its claims.
2.	Exploration of Failure Cases: The learned, fixed parent-child relationships in the tree might be a limitation for scenes with very complex, non-rigid deformations involving topological changes (e.g., cloth tearing, liquids splashing). The paper would be more complete if it discussed these potential failure cases where the assumed motion hierarchy might break down.

---

> ### Author Rebuttal · Authors · 2025-07-30
>
> Thank you for your insightful and positive comments! Below, we provide a point-by-point response to address your concerns. We welcome further discussion to improve the clarity and effectiveness of our work.
>
> > **W1 & Q3: Analysis of constructed motion trees.**
>
> We appreciate the reviewer’s suggestion to provide more insight into the learned motion forest structure. To address this, we have conducted a quantitative analysis of the distribution of tree depths across representative scenes from the DNeRF and Neural 3D Video datasets. In the following table, tree depths are divided into six bins. For each interval, the number of trees (**#Tree**) and the average number of Gaussians per tree (**#Gauss**) are reported across four scenes. This analysis reveals a long-tailed distribution: over 97% of motion trees have depths within 10, while only a few reach deeper hierarchies. This supports the claim that most Gaussians are captured through relatively shallow hierarchical motion patterns, with deeper structures emerging only where necessary.
>
> |      Scene       |  Depth   | #Tree | #Gauss |   Scene    |  Depth   | #Tree | #Gauss |
> | :--------------: | :------: | :---: | :----: | :--------: | :------: | :---: | :----: |
> | Cut Roasted Beef |  [1, 5]  | 10191 |  7.4   | Sear Steak |  [1, 5]  | 10572 |  7.9   |
> |                  | [6, 10]  | 1530  |  48.0  |            | [6, 10]  | 1482  |  40.9  |
> |                  | [11, 15] |  252  | 131.8  |            | [11, 15] |  208  | 196.7  |
> |                  | [16, 20] |  44   | 269.7  |            | [16, 20] |  37   | 190.1  |
> |                  | [21, 25] |   7   |  40.9  |            | [21, 25] |   4   |  61.0  |
> |                  | [26, 29] |   1   |  34.0  |            | [26, 29] |   0   |  0.0   |
> |   Hellwarrior    |  [1, 5]  |  529  |  4.9   |   Mutant   |  [1, 5]  |  639  |  9.7   |
> |                  | [6, 10]  |  159  |  36.3  |            | [6, 10]  |  279  |  83.6  |
> |                  | [11, 15] |  32   | 222.5  |            | [11, 15] |  89   | 367.7  |
> |                  | [16, 20] |   7   | 160.0  |            | [16, 20] |  16   | 321.0  |
> |                  | [21, 25] |   2   | 928.5  |            | [21, 25] |   2   | 328.0  |
> |                  | [26, 29] |   1   | 597.0  |            | [26, 29] |   0   |  0.0   |
>
> Additionally, we visualized Gaussian trajectories on the Neural 3D Video dataset by coloring all Gaussians bound to the same motion tree with same color. The results reveal that coherent motion trees consistently emerge around moving objects. For instance, in the flame_salmon scene, a motion tree clearly corresponds to the motion of a hand, with its associated Gaussians localized around the hand. In contrast, the static background is typically composed of a large number of isolated Gaussians without parent node (i.e., node level = 0). This behavior arises naturally from our tree growth mechanism, which is coupled with Gaussian densification and driven purely by gradients from reconstruction loss. Since moving regions introduce higher pixel inconsistencies across time, their corresponding gradients tend to be larger, promoting more active tree growth and hierarchical modeling in dynamic regions.
>
> While figures are not able to be included figures in the rebuttal, we will incorporate both the quantitative statistics and qualitative visualizations in the revision to better illustrate the structure and interpretability of the learnt motion trees.
>
> > **W2: Exploration of Failure Cases.**
>
> We thank the reviewer for pointing out this limitation. Indeed, our current formulation may struggle to fully capture topological changes or complex deformations such as cloth tearing or liquid splashing. These cases involve discontinuous motion patterns that challenge the assumption of temporally coherent, hierarchical motion. While our framework is effective for a wide range of objects and scenes with structured dynamics (as demonstrated in the benchmark datasets), we acknowledge that future extensions could explore more complex  tree topologies, or integrate additional cues such as segmentation or optical flow to better handle such extreme non-rigid phenomena. We will clarify this limitation in the revision.
>
> > **Q1: Discussion on potential for user interaction.**
>
> We appreciate the reviewer’s insightful suggestion. We believe the contribution of TreeSplat is complementary to methods like SC-GS. SC-GS models the motion of static Gaussians via external control points and an implicit MLP, where the trajectory of each Gaussian is interpolated from nearby control points using a linear blend skinning [1]. It also incorporates an adaptive densification strategy for these control points. Points with high gradients is cloned to better capture complex motion.
>
> Given this setup, TreeSplat can be naturally integrated into SC-GS by modeling hierarchical motion relationships among these control points. In such a framework, manipulating a high-level node (e.g., a learned "shoulder" node) could  propagate intuitive transformations to its descendants (e.g., "arm" Gaussians), enabling hierarchical editing or animation. Although not the primary focus of this work, we believe this integration offers a promising direction for future research in interactive 3D scene control. This discussion of the potential for user control will be included in the revision to better contextualize our contribution.
>
> [1] "Embedded deformation for shape manipulation." ACM SIGGRAPH 2007
>
> > **Q2: Robustness to Topological Changes & Role of Tree Growth.**
>
> We appreciate the reviewer’s thoughtful question. Our current framework indeed models each Gaussian’s motion via a motion tree with a one-to-one binding between Gaussian primitives and leaf motion nodes. The tree growth, including Leaf Expansion and Depth Promotion mechanisms, are primarily designed to progressively refine and structure the motion hierarchy, capturing multi-scale, correlated motions across the scene. However, these mechanisms do not explicitly detect or adapt to topological changes such as objects breaking apart. In scenarios involving significant discontinuities or topological shifts, the fixed hierarchical relationship may not perfectly capture the underlying motion patterns. While such failure modes are rare in our benchmark datasets, we acknowledge this as a limitation and will clarify it in the revision.

---

> > ### Comment · Reviewer_QfP4 · 2025-08-05
> >
> > The authors address most of my concerns, I also check new raised question from other reviewers and related responses.
> > The main novelty of this work is about motion hierarchy modeling for GS, it is current hot topic for GS reprenstation for dynamic objects. I agree this work contribute a key idea for this direction, some follow-up 'better' strategies are expected and acheive better rendering results.

---

> > > ### Author Response · Authors · 2025-08-05
> > >
> > > Dear Reviewer QfP4,
> > >
> > > Thank you for your encouraging follow-up comment. We are pleased to learn that the additional clarifications have addressed most of your concerns.
> > >
> > > We appreciate your recognition that the proposed motion hierarchy provides a key idea for dynamic Gaussian Splatting. We share the view that additional strategies built upon our framework can be explored in future work to enhance rendering quality.
> > >
> > > If you have any further questions or additional points you’d like us to address, please don’t hesitate to let us know.
> > >
> > > Best regards,
> > >
> > > The Authors of Submission 2287

---

### Official Review · Reviewer_iFUQ · 2025-07-02

**Clarity:** 3
**Significance:** 3
**Originality:** 3
**Rating:** 4
**Confidence:** 4

**Summary:**

This paper proposed a dynamic Gaussian splatting method which adopted a tree data structure to model the dependent motion. The deformation of each Gaussian primitive is computed through traversing the ancestral path, and make the motion of Gaussian primitives to be connected and not independent. Experiments on several popular datasets show the effectiveness.

**Questions:**

Please refer to the Weaknesses.

**Ethical Concerns:**

["NO or VERY MINOR ethics concerns only"]

**Final Justification:**

After reading the response of authors and other reviewers' comments, I think the response solved some of my concerns and I decide to keep my original positive score. And I hope the author can add the necessary citation and comparison with [1].

[1] 4d gaussian splatting with scale-aware residual field and adaptive optimization for real-time rendering of temporally complex dynamic scenes, ACMMM2024 Best Paper Candidate.

**Limitations:**

Yes

**Paper Formatting Concerns:**

Fig. 2 needs to be centered.

**Quality:**

3

**Strengths And Weaknesses:**

**Strengths**

1. It's reasonable that the motion or adjacent primitives is connected, and this paper designed a tree structure to model the independent motion of Gaussian primitives.
2. Experiments on several datasets also show that this method has great trade-off between quality and efficiency.

**Weaknesses**

1. Some implementations or designs are not explained clearly. How to decide whether apply the "Leaf Expansion" and "Depth Promotion" operations? Is there any factor like the densification threshold?

2. Missing some ablation studies like the threshold of "Leaf Expansion" and "Depth Promotion" operations.

3. How to make the connected Gaussian primitives to be constructed in a motion tree if only dependent on the "Leaf Expansion" and "Depth Promotion" operations?

4. In the initialization stage, each Gaussian primitive is bound to a single motion tree. And maybe it's not reasonable because there are some SFM points are connected.

5. How to avoid the situation that the dependent Gaussian primitives are connected in a motion tree?

---

> ### Author Rebuttal · Authors · 2025-07-30
>
> Thank you for your insightful comments! Below, we provide a point-by-point response to address your concerns. We welcome further discussion to improve the clarity and effectiveness of our work.
>
> > **W1-1: When to apply Leaf Expansion and Depth Promotion ?**
>
> It is great that reviewer is asking! *Leaf Expansion* and *Depth Promotion* are applied in an alternating manner at each Gaussian densification step during training.  Following the original 3D Gaussian Splatting setup, Gaussian densification is performed every 100 iterations starting from the 500th iterations. At each densification step, we alternate the two operations according to a fixed schedule: for every 5 densification steps, we apply *Leaf Expansion* in the first 4 steps, followed by *Depth Promotion* in the 5th step. This simple strategy empirically balances the width and depth of the motion tree and leads to effective hierarchical motion modeling.  Details about when to apply these two operations are elaborated from line 261 to line 264 in the manuscript, we will make it clearer in revision.
>
> > **W1-2: Other factors in tree construction like densification threshold.**
>
> Thanks! In our method, we do not introduce additional hyperparameters for tree growth. It is triggered automatically along the Gaussian densification procedure. To further clarify, we will revise the paper to explicitly mention this.
>
> > **W2: Threshold ablation of Leaf expansion and Depth promotion operations.**
>
> We appreciate the reviewer’s interest in the details of tree growth! As stated earlier, there isn't a single threshold for growth that we can ablate. But indeed, we do control the growth process by tuning the schedule of different operations. In Section 5.3, we conduct an ablation (Effect of Tree Depth) by varying this interval of *Depth Promotion* relative to *Leaf Expansion*. The best performance is achieved when *Depth Promotion* is applied after every 4 steps of *Leaf Expansion*, which balances tree width and depth. We will clarify this coupling and highlight the ablation results more explicitly in the revision.
>
> > **W3: How to make the connected Gaussian primitives constructed in a motion tree if only dependent on the "Leaf Expansion" and "Depth Promotion" operations?**
>
> Thank you for the question! The tree grows in conjunction with Gaussian densification, where new Gaussians are generated through either cloning or splitting from existing ones. These newly generated Gaussians are spatially adjacent to their source Gaussians, and our Leaf Expansion and Depth Promotion operations explicitly link them under in the tree as parent-child. This ensures that Gaussians with spatial and temporal proximity are hierarchically connected within the same motion tree. Over time, motion trees naturally group structurally related Gaussians, thereby enabling collaborative motion modeling across connected primitives.
>
> > **W4: In the initialization stage, each Gaussian primitive is bound to a single motion tree. And maybe it's not reasonable because there are some SFM points are connected.**
>
> Thank you for the thoughtful observation. We agree that Gaussians reconstructed from SfM may exhibit spatial or structural connections. In our framework, each Gaussian is bound to a single motion tree throughout training. This design choice stems from the difficulty of reliably inferring motion dependencies from sparse SfM points, where inaccurate correlations could introduce significant noise and hinder reconstruction. To empirically validate this, we conducted an ablation on the DNeRF dataset: we partitioned the initialization volume ($[-1.2, 1.2]^3$) into a $16 \times 16 \times 16$ voxel grid, and assigned each Gaussian to a tree based on its voxel location. This voxel-based pre-grouping introduces hardcoded spatial correlations. As a result, the average PSNR dropped from 37.11 dB (default) to 35.89 dB. This demonstrates that enforcing spatial proximity as motion correlation can be detrimental. Our method instead learns motion tree structure progressively in conjunction with Gaussian densification, which proves to be more robust and effective in practice.
>
> > **W5: How to avoid the situation that the dependent Gaussian primitives are connected in a motion tree?**
>
> Thank you for raising this concern. Actually, our design intentionally embraces spatial dependency among Gaussians, rather than avoiding it. The core motivation of our motion tree is to capture and exploit such correlations through hierarchical structure. During densification, new Gaussians are created in local neighborhoods, and their motion branches are extended from existing parent nodes, naturally preserving spatial continuity. Thus, Gaussians that are spatially related are encouraged to share motion components through shared ancestry in the tree. This promotes coherent motion modeling rather than viewing such dependency as a conflict.

---

> ### Comment · Reviewer_iFUQ · 2025-08-07
>
> Thanks for authors' response. After reading the response of authors and other reviewers' comments, I think the response solved most of my concerns and I decide to keep my original positive score. And I hope the author can add the necessary citation and comparison with [1]. The implementation details should also be further refined in the revised manuscript.
>
> [1] 4d gaussian splatting with scale-aware residual field and adaptive optimization for real-time rendering of temporally complex dynamic scenes, ACMMM2024 Best Paper Candidate.

---

> > ### Author Response · Authors · 2025-08-07
> >
> > Dear Reviewer iFUQ,
> >
> > Thank you very much for your encouraging follow up and for confirming that our rebuttal has resolved most of your concerns. We greatly appreciate your continued support and positive evaluation.
> >
> > We will incorporate a proper citation of reference [1] and provide a comparison with that method in the revised manuscript. We will also refine the implementation details to ensure greater clarity and reproducibility.
> >
> > Best regards,
> >
> > The Authors of Submission 2287

---

### Comment · Area_Chair_XSvv · 2025-08-04

Dear reviewers,

The authors' rebuttal has been uploaded. Could you please read the responses carefully and check if there are any questions or concerns that you would like to further discuss with the authors?

The author-reviewer discussion phase is going to end soon this week (Aug 6 11:59pm AoE).

Best,

Your AC

---

### Note · Authors · 2025-08-14

Dear Program Chair, Senior Area Chair, Area Chair and Reviewers,

We sincerely thank all reviewers for the constructive and thoughtful comments:

- Reasonable and Elegant Technical Design (Reviewer iFUQ, gmM8, QfP4)
- Novel motion hierarchy modeling (Reviewer QfP4)
- Outstanding results with extremely fast rendering speed (Reviewer iFUQ, QfP4)

We truly appreciate all reviewers mentioned we have addressed most concerns and raised questions. Below, we summarize common points that the reviewers were concern about during the rebuttal and discussion phase:

- **Deeper analysis and statistics of the motion tree.**  Reviewers were commonly interested in the depth statistics and internal structure of the motion trees. We added quantitative tree depth tables and conducted trajectory visualizations to clarify these points. Results show that most Gaussians are organized through relatively shallow hierarchical motion patterns, with deeper structures emerging only when necessary.
- **Computational and memory overhead.**  Multiple reviewers requested concrete numbers on tree merging latency and GPU memory usage. We clarified that training time increases by only three minutes on the Neural 3D Video dataset, and extra memory overhead is limited to 200MB on DNeRF and 850MB on Neural 3D Video. Tree merging adds only about 1ms per scene, confirming the efficiency of our method.
- **Comparisons with heuristic hierarchy designs.**  Reviewers asked for side by side results against heuristic semantic trees, fixed three layer trees, and related dynamic Gaussian Splatting variants.  We re-implemented these baselines and demonstrated consistent gains of up to two decibels PSNR and comparable SSIM.
- **Additional Visual quality Comparison.**  Reviewers requested video comparisons and additional metrics such as SSIM and LPIPS. We have supplied full metric tables on the Neural 3D Video dataset. We will provide side-by-side video comparisons in the supplementary material and release our code with raw results for transparency and reproducibility.

Finally, we are deeply grateful for the reviewers’ thoughtful engagement throughout the rebuttal. We hope the additional analyses and newly added experiments have helped clarify the remaining concerns and will be taken into consideration during the final assessment.

Best regards,

The Authors of Submission 2287

---

### Decision · Program_Chairs · 2025-09-17

**Decision:**

Accept (poster)

**Comment:**

The paper introduces a dynamic gaussian splatting method that uses hierarchical motion tree for coarse-to-fine motion representation. For high performance, the method also proposes a pre-aggregation strategy that liminates traversal overhead. The method achieves better rendering reconstruction accuracy with competitive runtime performance (eg, training time, FPS, storage)

In the initial reviews, reviewers raised questions and concerns on:
- Carity (iFUQ), implementation details (iFUQ), ablation study (iFUQ)
- Analyses on learned motion tree sturucture (QfP4)
- Limitation (QfP4)
- Further explanation on visual results (gmM8)
- Detail mechanism in the tree structure (wnrg)
- Difference from existing method [48] (wnrg)

After the discussion phase, most of the concerns are resolved. All the reviewers give **positive ratings (3 Borderline accept, Accept)**. AC recommends **accepting the paper**. It is highly recommend to incorporate all these feedbacks and discussions with reviewers in the final version.